# Flexible Curcumin-Loaded Zn-MOF Hydrogel for Long-Term Drug Release and Antibacterial Activities

**DOI:** 10.3390/ijms241411439

**Published:** 2023-07-14

**Authors:** Jiaxin Li, Yachao Yan, Yingzhi Chen, Qinglin Fang, Muhammad Irfan Hussain, Lu-Ning Wang

**Affiliations:** 1School of Materials Science and Engineering, University of Science and Technology Beijing, Beijing 100083, China; 2School of Shunde Graduate, University of Science and Technology Beijing, Foshan 528399, China

**Keywords:** curcumin, MOF, sodium alginate hydrogel, drug release, antibacterial activity

## Abstract

Management of chronic inflammation and wounds has always been a key issue in the pharmaceutical and healthcare sectors. Curcumin (CCM) is an active ingredient extracted from turmeric rhizomes with antioxidant, anti-inflammatory, and antibacterial activities, thus showing significant effectiveness toward wound healing. However, its shortcomings, such as poor water solubility, poor chemical stability, and fast metabolic rate, limit its bioavailability and long-term use. In this context, hydrogels appear to be a versatile matrix for carrying and stabilizing drugs due to their biomimetic structure, soft porous microarchitecture, and favorable biomechanical properties. The drug loading/releasing efficiencies can also be controlled via using highly crystalline and porous metal-organic frameworks (MOFs). Herein, a flexible hydrogel composed of a sodium alginate (SA) matrix and CCM-loaded MOFs was constructed for long-term drug release and antibacterial activity. The morphology and physicochemical properties of composite hydrogels were analyzed by scanning electron microscopy (SEM), Fourier transform infrared spectroscopy (FT-IR), X-ray diffraction (XRD), ultraviolet-visible spectroscopy (UV-Vis), Raman spectroscopy, and mechanical property tests. The results showed that the composite hydrogel was highly twistable and bendable to comply with human skin mechanically. The as-prepared hydrogel could capture efficient CCM for slow drug release and effectively kill bacteria. Therefore, such composite hydrogel is expected to provide a new management system for chronic wound dressings.

## 1. Introduction

Chronic wounds and the accompanying persistent bacterial inflammation have always been a vital challenge in clinical treatment. For the treatment of wounds, curcumin (CCM) is highly recognized for benefitting human health [1]. It is a natural lipophilic polyphenol and natural antioxidant compound that has been widely used as a food additive, food coloring [2], and flavoring agent [3] in the past decades, and in recent years researchers have revealed the pleiotropic nature of the biological effects of this molecule. Much research has clarified its antioxidant, anti-inflammatory, and antibacterial activities, which are beneficial for wound healing [4]. However, poor water solubility (under acidic and neutral conditions), chemical instability (especially under neutral and alkaline conditions), rapid-metabolic-rate-limited bioavailability, and delivery efficiency severely affect its application in wound healing [5]. In this regard, nanotechnology-based delivery systems (encapsulating the drug into specific nanocarriers) are considered a promising technology for circumventing these obstacles.

To date, various drug delivery platforms have been developed, including polymeric nanoparticles [6], carbon-based nanostructures, lipid-based nanoparticles [7], and inorganic nanostructures [8]. Among them, carbon-based nanostructures display high surface area, functionalization versatility, and drug-loading capacity for drug delivery [9]. Biomolecules such as lipids or proteins are good choices as biocompatible platforms [10]. Inorganic or organic nanoparticles are appealing for their altered pharmacokinetics and biodistribution profiles, sometimes presenting unique photonic/thermal/electronic/magnetic effects in guiding the accumulation or release of drugs [11,12]. To combine the merits of loading efficiency, tunability, and biocompatibility, metal-organic frameworks (MOFs) stand out as promising antibacterial platforms. MOFs are a class of highly crystalline and nanoporous materials that can be built from various metal-ion metal-clusters and organic linkers for tunable chemical and topological structures [13]. Recently, increasing use of MOFs has been made in drug storage and release for their combined properties of high adsorption capacity, tunable host-guest interaction and release efficiency, biocompatibility, and nontoxicity [14].

To be properly used as a wound dressing, it should also be able to fill wound spaces and provide mechanical stability as well as good penetrability to water vapor and metabolites. Hydrogels with three-dimensional (3D) hydrophilic polymeric structures can meet these requirements [15] by mimicking the native extracellular matrix (flexible and stable geometry, high plasticity, and not sticking to the wound) and creating a moist microenvironment for wounds [16]. Simultaneously, hydrogels can host various bioactive ingredients [17], and their high porosity and controllable crosslinking allow these ingredients to be delivered suitably for accelerating wound healing [18]. Hydrogels’ water absorption and penetrability arise from their flexible polymer network with many hydrophilic groups [19] (e.g., R-COOH, R-CONH_2_, R-NH_2_, R-OH, R-SO_3_H). These functional groups in hydrogels facilitate the adsorption of metal ions by binding with various oxygen-, nitrogen-, or sulfur-containing functional groups [20]. This also provides anchoring sites for the nucleation and growth of MOFs, highlighting the rational design of an efficient wound-dressing system.

Numerous studies have been conducted to combine the above components into complexes for wound healing. As for the typical binary system, (1) functional MOFs carrying drugs have excellent antibacterial properties via the controllable release of antibacterial drugs and metal ions for efficient wound healing [21,22]. Recent research has suggested that modifying the surface of MOFs could enhance their adherence to bacterial cells, inhibiting the growth of bacteria [23]. However, it is in powder, making it challenging to adhere to the skin stably. (2) MOF-anchored hydrogel dressings are an alternative that could achieve antibacterial action by destroying the integrity of bacterial cell membranes and decomposing H_2_O_2_ to improve wound hypoxia due to the peroxidase performance of MOF [24,25,26]. However, without drug loading, antimicrobial efficiency is limited. (3) The drug-included hydrogel delivery system, which encapsulates drugs physically, has excellent deformability and oxygen transmission rates as wound dressings [27,28]. However, it is difficult to regulate the drug release rate. Recently, several studies have demonstrated the excellent antimicrobial activity of drug–MOF–biomolecular carrier systems (ternary systems) [29,30,31]. Despite the improved performance for wound healing, its preparation process was relatively complicated. This is because synthesizing the precursors requires long preparation cycles, harsh preparation conditions, and cumbersome preparation processes. Therefore, simplifying the preparation methods for ternary systems is currently an issue that needs to be addressed.

In this study, CCM ligands and Zn^2+^-centered MOFs with a similar structure to the zeolitic imidazolate framework (ZIF-8) were facilely prepared on sodium alginate (SA) hydrogel to improve the stability and administration efficiency of CCM, affording controlled release of CCM and improved bioavailability. Zn^2+^, introduced as a cross-linking agent and presented on the surface of the prepared hydrogels, could also be used as a metal-ligand center linked to both 2-methylimidazole (2-mIM) ligand and CCM, resulting in a one-step synthesis of CCM-loaded Zn-MOF hydrogels (CCM@ZIF-8@SA), which simplified preparation process to a great extent. The composite hydrogel displayed controlled antibacterial and anti-inflammatory effects by changing the amount of hosted CCM. The composite hydrogel also exhibited reasonable flexibility that could be twisted and bent on human skin, which is promising for use as a wound dressing. Such a system fully uses the performance advantages and structural characteristics of each conventional single component, circumventing the shortcomings of CCM, traditional dressings, and MOFs in biomedical applications, thus providing a rational design guideline for an antibacterial CCM-based platform for wound healing.

## 2. Results and Discussion

### 2.1. Characterization of CCM@ZIF-8@SA-Composite Hydrogels

Zn^2+^ was used as the cross-linking agent for the preparation of flexible sodium SA hydrogels, and the surficial Zn^2+^ on the as-prepared hydrogel could also work as a metal-coordinated center to simultaneously connect with the 2-mIM ligand and CCM, leading to the formation of CCM@ZIF-8@SA, as shown in Figure 1a. With increasing CCM amounts (1.0, 3.0, and 5.0 mg), three kinds of composite hydrogels were obtained, 1CCM@ZIF-8@SA, 3CCM@ZIF-8@SA, and 5CCM@ZIF-8@SA, respectively. Optical and scanning electron microscopy (SEM) images of the as-prepared SA, ZIF-8@SA, and 1/3/5CCM@ZIF-8@SA were depicted in Figure 1. The optical images showed that the SA hydrogel after Zn^2+^-crosslinking was transparent (Figure 1b), and after loading the ZIF-8 particles, the sample changed from transparent to white (Figure 1c). When CCM was added together with the coordination ligand, the sample color turned orange and gradually deepened as the CCM amount increased (Figure 1d–f). As shown in the SEM image (Figure 1g–k), they all had a porous sponge structure, which contributed to water retention/penetration and sustained drug release inside the hydrogel. As shown in the magnified insets in Figure 1g,h, the surface of SA hydrogel was smooth, while coordination of Zn^2+^ with 2-mIM gave rise to regular dodecahedral ZIF-8 particles homogeneously dispersed on SA (ZIF-8@SA). From Figure 1i, it can be seen that a small dosage of CCM did not change the surface state slightly, but when the added CCM amount was increased to 3 mg, most of the regular dodecahedral structure was replaced by a sheet structure (Figure 1j), meaning the CCM could work as the regulator in the coordination reaction and even as another ligand to coordinate with Zn^2+^, thus rebuilding the MOF structure [32,33]. With CCM increasing to 5 mg (Figure 1k), the surface dodecahedral structures were changed to lamellar structures, verifying the similar role of CCM as ligands. The lamellar structures are expected to serve as effective channels for drug release.

The X-ray diffraction (XRD) patterns (Figure 2a) showed that ZIF-8 exhibited characteristic diffraction peaks at 7.39, 10.45, 12.84, and 16.51°, which corresponded to the (110), (200), (211), and (222) planes, respectively [34]. These typical peaks appeared in ZIF-8@SA and continued to endure in the 5CCM@ZIF-8@SA (containing the most CCM) samples, indicating that ZIF-8 crystals were successfully formed onto the hydrogel [35], and the participation of CCM did not affect the packing of ZIF-8. The Fourier transform infrared spectroscopy (FT-IR) spectra are shown in Figure 2b. The samples displayed distinct absorption peaks at 3435, 2927, 1620, and 1567 cm^−1^. The peaks at 2927 and 1567 cm^−1^ were attributed to the stretching vibrations of the C-H and C-N bonds of the aliphatic group in 2-methylimidazole. In the spectrum of 1/3/5CCM@ZIF-8@SA, the peak at 1620 cm^−1^ corresponded to the superposition vibrations of the C=C and C=O bonds in CCM. Pure CCM exhibited a characteristic absorption peak (phenolic O-H stretching vibration) at 3490 cm^−1^. Still, this peak was blue-shifted in the 3490–3435 cm^−1^ band in 1/3/5CCM@ZIF-8@SA, indicating that the hydrogel surface was gradually covered by CCM [36]. UV-Vis absorption spectra were provided to analyze the composition. As shown in Figure 2c, bare ZIF-8 has a characteristic absorption peak at 226 nm, which also appeared in 1/3/5CCM@ZIF-8@SA, indicating that coating of ZIF-8 on the SA surface. Similarly, the peak at approximately 427 nm, corresponding to CCM, was also found in 1/3/5CCM@ZIF-8@SA. This indicated that CCM had been successfully loaded into the ZIF-8 framework. In contrast to pure CCM, the composite showed a redshift of approximately 13 nm. This further revealed the strong interaction between CCM and Zn^2+^ in either ZIF-8 or the hydrogel, which decreased the band gap between the π−π * [37] electronic transition of CCM. It is supposed that CCM contains highly conjugated 1, 3-diketone moieties (1, 3-diketones and two enols) in the tautomer, which could be connected with Zn^2+^ to form porous skeleton compounds with stable structures. In the Raman spectra (Figure 2d), no obvious bond was found in SA; ZIF-8@SA showed the characteristic bonds of ZIF-8 at 286 cm^−1^ (Zn-N vibrations in the ZnN_4_ tetrahedron) [38], 1123 cm^−1^ (C-N stretching), and 1461 cm^−1^ (C-H) [39], indicating the successful loading of ZIF-8 onto SA; the extracted bending vibration of the imidazole ring at 694 cm^−1^ was assigned to Zn-N vibrations [40]. These distinct Raman bands of CCM were also observed in 1/3/5CCM@ZIF-8@SA, further confirming the incorporation of CCM [41].

Based on the thermogravimetric (TGA) analysis, the thermal stability of the sample was determined. It has been revealed that at elevated temperatures, CCM begins to degrade weightlessly at 200 °C, and ZIF-8 starts to collapse weightlessly and structurally at 580 °C [41]. As shown in Figure 3a, the weightlessness curve of SA exhibited a stable decreasing trend, but ZIF-8@SA experienced a rapid weight loss at 544 °C caused by the collapse and degradation of the ZIF-8 frame loaded on the SA. Meanwhile, the 1/3/5 CCM@ZIF-8@SA samples underwent rapid weight loss at 243 and 544 °C. The first prompt weight loss that occurred at 243 °C was due to the degradation of the encapsulated CCM. The increase in weight loss temperature compared to the pure CCM weight loss temperature was caused by the interaction of the CCM with the metal part in the ZIF-8 framework. And the collapse degradation of the ZIF-8 framework triggered the second rapid weightlessness at 544 °C. The pore structure parameters of the sample as determined by the BET test, in Figure 3b, ZIF-8 and 1/3/5CCM@ZIF-8 exhibited I-type adsorption curve, with significant increases in N_2_ uptake at low relative pressure (<0.01 MPa) due to the presence of micropores. As seen in Table 1, the specific surface area of 1/3/5CCM@ZIF-8 was decreased by 1.63%, 1.98%, and 9.51%, the pore volume was reduced by 3.2%, 15.8%, and 31.6%, and the average pore size was also reduced by 11.9%, 12.3%, and 14.8%, respectively, while the pure ZIF-8 had the specific surface area of 968.3 m^2^/g, and the pore volume and pore size were 0.461 cm^3^/g and 3.302 nm. The decrease in the porosity of 1/3/5CCM@ZIF-8 was ascribed to the encapsulating of CCM into the ZIF-8 framework. The pore size distributions of ZIF-8 and 1/3/5CCM@ZIF-8 (Figure 3c) were mainly concentrated in the microporous range, and the pore size distribution of ZIF-8 and 1/3/5CCM@ZIF-8 is mainly in the microporous range (Figure 3c). However, the average pore size was mesoporous, which may have been caused by the gap error between particles in the test.

The mechanical properties of composite hydrogels were also tested. As shown in Figure 4a, the tensile strength increased by adding CCM and ZIF-8. Compared with the SA hydrogel (7.66 MPa), the ultimate tensile strengths of the ZIF-8@SA and 5CCM@ZIF-8@SA were 10.67 MPa and 11.95 MPa, respectively. This might be attributed to the formation of supplementary chemical cross-linking (i.e., coordination bond, hydrogen bond) between ZIF-8 and SA [42,43]. Bending and twisting tests were conducted on 5CCM@ZIF-8@SA samples to verify the flexibility. As shown in Figure 4b, the 5CCM@ZIF-8@SA sample exhibits significant distortion and maintains the original shape after twisting. When the hydrogel adhered to the finger joints and with bending (Figure 4c), the sample did not break after bending at 120°, 90°, and 60°. These results indicate its good toughness and flexibility, which are promising for use as a wound dressing.

### 2.2. Drug Release Behavior of CCM@ZIF-8@SA-Composite Hydrogel

As previously reported, encapsulated CCM is relatively stable against degradation [44] under neutral and alkaline conditions. As a result, the release of CCM in 1/3/5CCM@ZIF-8@SA in 0.01M PBS and Tween 20 (0.5 *v*/*v*%) (PH = 7.4) was investigated. From the quantitative analysis of turmeric quality of 1/3/5CCM@ZIF-8@SA samples, shown in Table 2, the encapsulation efficiency values of 1/3/5CCM@ZIF-8@SA CCM were 50%, 56.2%, and 76.7%, respectively. Based on the wavelength scans of CCM solutions with different concentrations of CCM and CCM standard curves, the relationship between CCM concentration and absorbance was obtained by linear fitting as y = 0.14617x + 0.02948, with a correlation coefficient R^2^ = 0.99562 (Figure 5a,b). The cumulative percentage of CCM released from the three samples of 1CCM@ZIF-8@SA, 3CCM@ZIF-8@SA, and 5CCM@ZIF-8@SA could be calculated from this linear relationship, as shown in Figure 5c. The release of CCM increased rapidly in the first 4 h, after which the rising release trend gradually slowed and finally approached a steady state. This is because CCM is encapsulated inside the tiny pores of the ZIF-8 framework. At the initial stage of release, the CCM on the surface and near the pores diffused into the solution (related to the fast swelling capacity of the sample surface) [45], whereas most of the CCM inside remained encapsulated inside the particles [46]. Therefore, the release of CCM inside the granules would be slow, and a long-term release behavior could be achieved. A similar release pattern was observed for three samples. Compared to the relatively rapid release of 1CCM@ZIF-8@SA (about 16.4% in 72 h), CCM release from 3CCM@ZIF-8@SA and 5CCM@ZIF-8@SA samples was controlled and significantly delayed (about 22.1% and 25.9% in 72 h), and most importantly, the duration of the sustained release of CCM for 5CCM@ZIF-8@SA could reach 72 h or above with a slower releasing rate. Due to its eutectic surface, 1CCM@ZIF-8@SA released fast, possibly because it would dissolve rapidly in large amounts upon touching the PBS containing Tween 20 (0.5 *v*/*v*%) solution upon contact. In contrast, most CCM molecules entered the interior of ZIF-8 for 5CCM@ZIF-8@SA and 3CCM@ZIF-8@SA in addition to the surficial few molecules. In particular, CCM in the 5CCM@ZIF-8@SA group fully integrated with the pores in ZIF-8 to achieve the optimal coating rate of the drug, delaying the release rate and prolonging the release time. Based on the SEM images, this conclusion is consistent with the observed morphologies and structures.

### 2.3. Antibacterial Activities of CCM@ZIF-8@SA-Composite Hydrogels

To assess the antibacterial activities of drug-loaded complex hydrogels, we selected Gram-positive *Staphylococcus aureus* (*S. aureus*, BNCC186335) and Gram-negative *Escherichia coli* (*E. coli*, BNCC336902) for antibacterial assays. Antibacterial performance is assessed by colony-forming unit (CFU) assay. After co-incubating the complex hydrogel with bacterial culture medium at 37.5 °C for 24 h, quantitative data (Figure 6a,c) and CFU assay (Figure 6b,d) showed that both the ZIF-8@SA and 1/3/5CCM@ZIF-8@SA samples showed good antimicrobial properties against both *S. aureus* and *E. coli*. Among them, ZIF-8@SA showed an antibacterial efficiency of 28% against *S. aureus* and 58% against *E. coli*. The antibacterial efficiencies against *S. aureus* increased to 30%, 42%, and 65% for 1/3/5CCM@ZIF-8@SA, respectively, 2–4 times higher than pure SA (18%). A similar trend was also found in killing *E. coli,* with values of 70%, 80%, and 86% for three samples. ZIF-8@SA performed better than bare SA, possibly because the release of Zn^2+^ from ZIF-8 could damage the bacteria. This may arise from the disruption of bacterial membrane permeability by Zn^2+^, which inhibits glycolysis, glucosyltransferase production, and polysaccharide synthesis in bacteria [47,48]. The antibacterial effects of the 1/3/5CCM@ZIF-8@SA hydrogels were thus the synergistic effect of CCM and Zn^2+^. Based on the coating results of the two types of bacteria and the quantitative analysis of OD values, the OD values followed a decreasing order of SA > ZIF-8@SA > 1CCM@ZIF-8@SA > 3CCM@ZIF-8@SA > 5CCM@ZIF-8@SA both for *E. coli* and *S. aureus*, with 5CCM@ZIF-8@SA performing the best. The excellent antibacterial properties of 1/3/5CCM@ZIF-8@SA highlight its practicality as a wound dressing.

The minimum bacteriostatic concentration (MIC) and minimum bactericidal concentration (MBC) were used to determine the access to the bacteriostatic ability of bacteriostatic substances that act on *E. coli* and *S. aureus* with SA, ZIF8@SA, and 1/3/5CCM@ZIF-8@SA. As shown in Figure 7a, the MIC values of pure SA against *E. coli* and *S. aureus* were 3.3 mg/mL and 5.3 mg/mL, indicating that the basal SA did not have good antibacterial properties. The MIC values of ZIF-8@SA against *E. coli* and *S. aureus* were 0.83 mg/mL and 1.67 mg/mL, which indicated that after loading ZIF-8, the hydrogel had a particular antibacterial ability. In the antibacterial experiment of 1/3/5 ZIF-8@CCM@SA samples, with the increase in CCM loading, the MIC value of *E. coli* gradually decreased to 0.67, 0.33, and 0.25 mg/mL. In addition, the MIC value of *S. aureus* gradually decreased to 0.83, 0.41, and 0.33 mg/mL. Similarly, the MBC results (Figure 7b) showed the same trend. The MBC values followed a decreasing order of SA > ZIF-8@SA > 1CCM@ZIF-8@SA > 3CCM@ZIF-8@SA > 5CCM@ZIF-8@SA both for E. coli and S. aureus, with 5CCM@ZIF-8@SA performing the best. Therefore, ZIF-8 could improve the stability and utilization efficiency of CCM and, finally, increase the bacteriostatic ability of CCM @ ZIF-8 @ SA.

## 3. Materials and Methods

### 3.1. Materials

2-mIM, CCM, SA, zinc chloride, and methanol (MeOH, for synthesis) were purchased from Beijing Bailinway Technology Co., Ltd., (Beijing, China), and phosphate buffer (0.01 M) was purchased from Beijing Lanyi Chemical Products Co., Ltd., (Beijing, China). Gram-positive *S. aureus* (BNCC186335) and Gram-negative *E. coli* (BNCC336902) were purchased from Beinong Yuhe Technology Development Co., Ltd., (Beijing, China).

### 3.2. Synthesis

(i) Preparation of SA

SA hydrogels were prepared using an ionic cross-linking method. First, sodium alginate (300 mg) was dissolved in water (10 mL) and thoroughly mixed in a surface dish to obtain aqueous sodium of solution alginate. A Petri dish containing 1 g of dissolved sodium alginate solution was weighed on an analytical balance before being transferred to the refrigerator frozen section for 24 h to solidify. ZnCl_2_ (545 mg) was dissolved in H_2_O (20 mL) and magnetically stirred to homogenize the solution. The freeze-fixed aqueous sodium alginate solution was mixed dropwise with 6 mL of the ZnCl_2_ solution to create the SA hydrogel. The reaction was allowed to run for approximately 2 h. Finally, the residual ZnCl_2_ solution on the hydrogel surface was washed with deionized water.

(ii) Preparation of ZIF-8@SA

ZIF-8@SA hydrogels were synthesized by co-precipitation at room temperature. 2-mIM (330 mg) was dissolved in a MeOH solution (10 mL), and the mixture was magnetically agitated (as described above) until it became clear. The solution above was then added to a 35 mm Petri plate with the SA hydrogel created in step (i), and the reaction was allowed to run for 4 h. After the reaction, the hydrogel was oven-dried at 60 °C for 8 h. Finally, a ZIF-8@SA hydrogel with a white solid attached to the surface was obtained.

(iii) Preparation of 1/3/5CCM@ZIF-8@SA

Similarly, CCM@ZIF-8@SA hydrogels were synthesized using a room-temperature co-precipitation method. 2-mIM (330 mg) and CCM (1.0, 3.0, and 5.0 mg) were dissolved in MeOH solution (10 mL), naming solution A/B/C, and magnetically stirred until the solution became clear. Then, the SA hydrogel prepared in step (i) was soaked in the above solution (in a 35 mm petri dish), and the reaction was allowed to continue for 4 h. The resulting hydrogel was dried in an oven at 60 °C for 8 h. Three types of hydrogels were obtained, all with CCM solids attached to their surfaces, and were named 1CCM@ZIF-8@SA, 3CCM@ZIF-8@SA, and 5CCM@ZIF-8@SA, respectively.

### 3.3. Characterizations

The analytical balance (OHAUS brand) used for weighing was purchased from Cangzhou Zhenqian Instrument Company (Cangzhou, China). The magnetic stirrer (IKA C-MAG HS 7) used for solution stirring was purchased from Beijing Chenshi Technology Trade Company (Beijing, China). The oven (DHG-9148A) used for sample drying was purchased from Shanghai Jinghong Laboratory Equipment Company (Shanghai, China). The optical images of the sample were recorded by the phone’s rear camera. The topography and microstructure were observed by the Regulus-8100 cryo-field emission scanning electron microscope with an acceleration voltage of 15 kV. Prior to the experiments, hydrogel samples were prepared. Due to the poor conductivity of the material, aside from using conductive glue to fix the water gel sample onto the experimental sample stage, pre-sputter gold treatment was also required. The XRD patterns were obtained on a Rigaku TTR3 X-ray diffractometer with a CuKα target (λ = 1.5 Å). The dried hydrogel samples were ground into powder before the experiment, and an appropriate amount of hydrogel powder was placed on a sample holder for scanning. The scanning range of 2θ was 5° to 40°, and the scanning speed was 10°/min. Raman scattering spectra were obtained on a micro-focus Raman spectrometer (inVia-Reflex) with an excitation wavelength of 785 nm. Similarly, the ground hydrogel powder was placed on a glass slide for the experiment. The scanning range of Raman spectra was 0–1800 cm^−1^, and the scanning step of 2 cm^−1^ was used to detect the crystal structure of the samples. The Cary 7000 UV-Vis diffuse reflectance spectrophotometer was used to test the hydrogel samples’ optical response range and intensity changes with a 200–700 nm scanning wavelength. Fourier transform infrared spectroscopy (Excalibur 3100) further detected the samples’ functional groups in the range of 500–4000 cm^−1^. The wet-state composite hydrogel was stretched at a 5 mm/min speed on a universal tensile testing machine (TA.HD PLUS), and the tensile strength was calculated by dividing the maximum stress by the area. In the drug release test, the release concentration of CCM was measured using a UV-Vis spectrophotometer (UV-2600).

### 3.4. Drug Release Tests

(i) The standard curve of CCM:

CCM (100 mg) was dissolved in 10 mL PBS and Tween 20 (0.5 *v*/*v*%) to prepare as the stock solution. Then, 1 mL was extracted from the above solution and diluted, and finally obtain a gradient of 10, 8, 6, 4, and 2 μg/mL of CCM solution. The UV-Vis absorption of these solutions was measured using a spectrophotometer in the wavelength range of 300–700 nm. The absorbance intensities at 425 nm were recorded for each concentration of the CCM solution.

(ii) CCM encapsulation efficiency

The total amount of CCM in 1/3/5CCM@ZIF-8@SA was obtained using the following method:

Calculated from the following equation:

Encapsulation efficiency (%) = $\frac{Loading}{Total\;amount\;of\;CCM}$ × 100%

Loading: the mass of 1/3/5CCM@ZIF-8@SA–the mass of ZIF-8@SA;

Total amount of CCM: the mass of curcumin in 6 mL solution A/B/C.

(iii) The drug release test of CCM@ZIF-8@SA-composite hydrogels:

The as-prepared CCM@ZIF-8@SA-composite hydrogel was soaked in 15 mL of PBS containing 0.5 *v*/*v*% Tween 20 (pH = 7.4). At specific time intervals (20 min, 40 min, 1, 2, 4, 8, 12, 14, 24, 36, 60, and 72 h), 3 mL of the soaking solution was analyzed. The absorbance intensities at 425 nm were measured for each sample. The concentration of CCM in each sample was determined by comparing the absorbance of the samples at 425 nm with the standard curve. Each experiment was performed three times in parallel, and the average results from the three experiments were used for data processing and analysis.

The 72 h release of CCM was calculated according to the following equation:

Release (%) = $\frac{Released\;amount\;of\;CCM}{Total\;amount\;of\;CCM}$ × 100%

### 3.5. Antimicrobial Properties

(i) Sample processing:

The as-prepared hydrogels were placed in a centrifuge tube and irradiated under an ultraviolet lamp for 30 min.

(ii) Medium configuring:

Briefly, 4 g of Luria–Bertani (LB) broth was added to 200 mL of deionized water and stirred to obtain a liquid medium, and 9.6 g of agar medium was added to 300 mL of deionized water and stirred thoroughly to obtain a solid medium. Then two different media were placed into the LX-B50L autoclave for sterilization (120 °C, 30 min). After the sterilization process was complete, the media were allowed to cool down. At last, the liquid medium was sealed with a sealing film and placed on the ultra-clean stage for future use. The solid medium was poured into 90 mm Petri dishes at a temperature of 70 °C. Approximately 10 mL of the medium was poured into each dish, and once it solidified, it was sealed with a sealing film and placed upside down on the ultra-clean stage for future use.

(iii) Bacterial resuscitation:

The strains used in this experiment are Gram-positive *S. aureus* (BNCC186335) and Gram-negative *E. coli* (BNCC336902). Before the experiment, 50 mL centrifuge tubes, respiratory membranes, and other experimental consumables were placed on a laminar flow hood and exposed to UV light for 30 min for sterilization. During the experiment, a small number of bacteria was taken with a heated inoculation loop and transferred into a 30 mL liquid medium. The culture was then sealed with a breathable membrane and incubated on a shaker at 37 °C for 18 h.

(iv) Bacterial proliferation:

Bacterial proliferation was analyzed by measuring the optical density (OD) value and the coating method. After 18 h of bacterial recovery, the bacterial solution was removed from the shaker and diluted thrice with a liquid culture medium. Then, 3 mL of the diluted bacterial solution was taken and placed in a cuvette, and the OD value of the bacteria was measured using a UV-vis spectrophotometer. The bacterial concentration was then diluted to 10^6^ CFU·mL^−1^ based on the OD value (each OD value = 10^8^ CFU·mL^−1^). The experimental group added 5 mL of bacterial liquid and a hydrogel sample to a 15 mL centrifuge tube. In contrast, in the control group, only 5 mL of bacterial liquid was added to a 15 mL centrifuge tube. Finally, the centrifuge tubes were sealed with a breathable membrane and placed in a constant temperature (37 °C) incubator for 24 h of cultivation.

(v) Determination of bacterial bacteriostatic rate:

After co-culturing for 24 h, the OD value at 600 nm was measured using a UV spectrophotometer. Next, the bacterial solution after co-culturing was diluted 10^3^, 10^4^, 10^5^, and 10^6^ times. Then 100 μL of different concentrations of the bacterial liquid was taken and coated onto the pre-prepared solid culture medium using spreaders. The culture plates were sealed with sealing film and transferred to a 37 °C incubator for cultivation. After 24 h of cultivation, the culture plates were taken out, and the number of bacterial colonies was observed and recorded using photography. The relative antibacterial rate was calculated according to the following formula:

Relative bacteriostatic rate (%) = $\frac{A-B}{A}$ × 100%

A—the OD values of the colonies in the blank group;

B—the OD values of the colonies in the experimental group.

## 4. Conclusions

In summary, a flexible CCM-loaded Zn-MOF hydrogel was facilely prepared by combining the drug with organic ligands through Zn^2+^ crosslinking and Zn^2+^ centered coordination. The results showed that CCM-loaded ZIF-8 crystals could be homogeneously synthesized onto SA hydrogel, with CCM becoming part of the ZIF-8 ligand. This would strengthen the encapsulation of CCM inside ZIF-8 frameworks and simultaneously enhance the crosslinking by the supplementary intermolecular forces. Consequently, flexible and tough composite hydrogels can be obtained to adapt to the mechanical behaviors of human skin. More importantly, this hierarchical structure could achieve long-term drug release and obtain exceptional antimicrobial properties against *E. coli* and *S. aureus*. In conclusion, such composite hydrogel promises a wound-healing system that can guide the future design of practical chronic wound dressings.

## Figures and Tables

**Figure 1 ijms-24-11439-f001:**
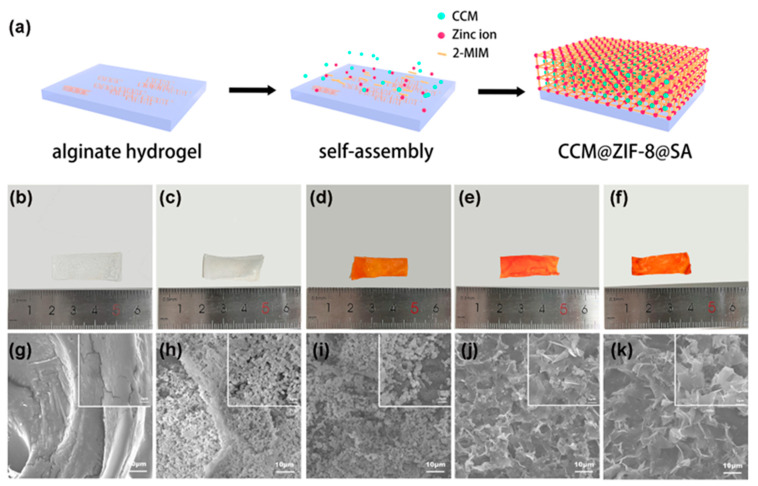
(**a**) Schematic diagram of the hydrogel synthesis. Optical images of SA (**b**), ZIF-8@SA (**c**), 1CCM@ZIF-8@SA (**d**), 3CCM@ZIF-8@SA (**e**), 5CCM@ZIF-8@SA (**f**). SEM images of SA (**g**), ZIF-8 @SA (**h**), 1CCM@ZIF-8@SA (**i**), 3CCM@ZIF-8@SA (**j**), 5CCM@ZIF-8@SA (**k**).

**Figure 2 ijms-24-11439-f002:**
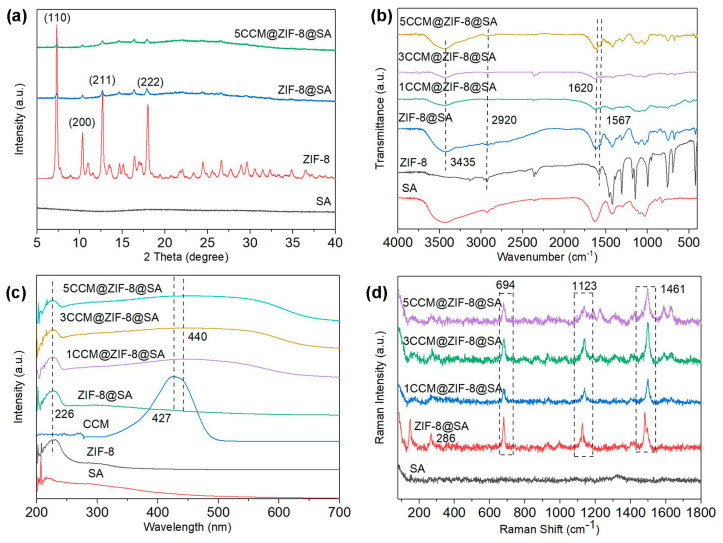
(**a**) XRD spectra of SA, ZIF-8, ZIF-8@SA and 5CCM@ZIF-8@SA. (**b**) FTIR spectra of CCM, ZIF-8, ZIF-8@SA, and 1/3/5CCM@ZIF-8@SA. (**c**) UV-Vis absorption spectra of SA, ZIF-8, ZIF-8@SA, and 1/3/5CCM@ZIF-8@SA. (**d**) Raman spectra of SA, ZIF-8@SA, 1/3/5CCM@ZIF-8@SA.

**Figure 3 ijms-24-11439-f003:**
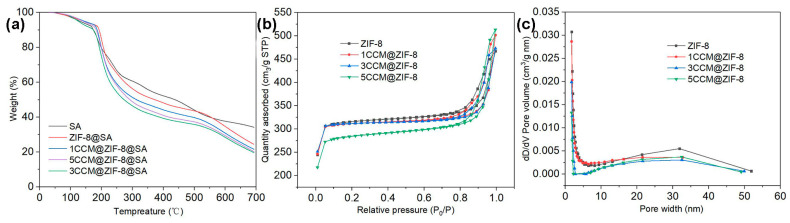
(**a**) TGA thermograms of SA, ZIF-8@SA,1/3/5CCM@ZIF-8@SA. (**b**) Linear absorption desorption isotherm of ZIF-8,1/3/5CCM@ZIF-8. (**c**) Pore size distribution of ZIF-8,1/3/5CCM@ZIF-8.

**Figure 4 ijms-24-11439-f004:**
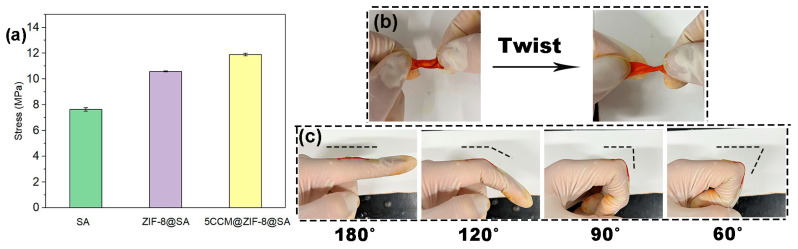
(**a**) Stress-strain image of SA, ZIF-8@SA, and 5CCM@ZIF-8@SA. (**b**) Twisting behavior of 5CCM@ZIF-8@SA. (**c**) Bending behavior of 5CCM@ZIF-8@SA.

**Figure 5 ijms-24-11439-f005:**
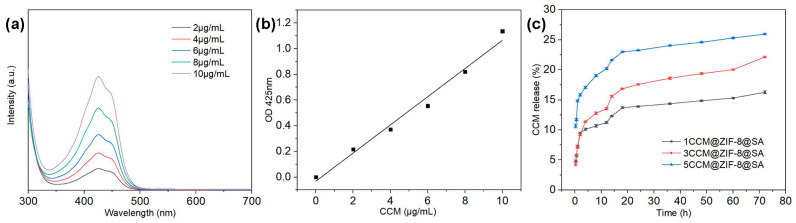
(**a**) Wavelength scans of ethanol solutions of different CCM concentrations, (**b**) Standard curve fitted to the actual concentration of CCM., (**c**) CCM released from CCM@ZIF-8@SA samples.

**Figure 6 ijms-24-11439-f006:**
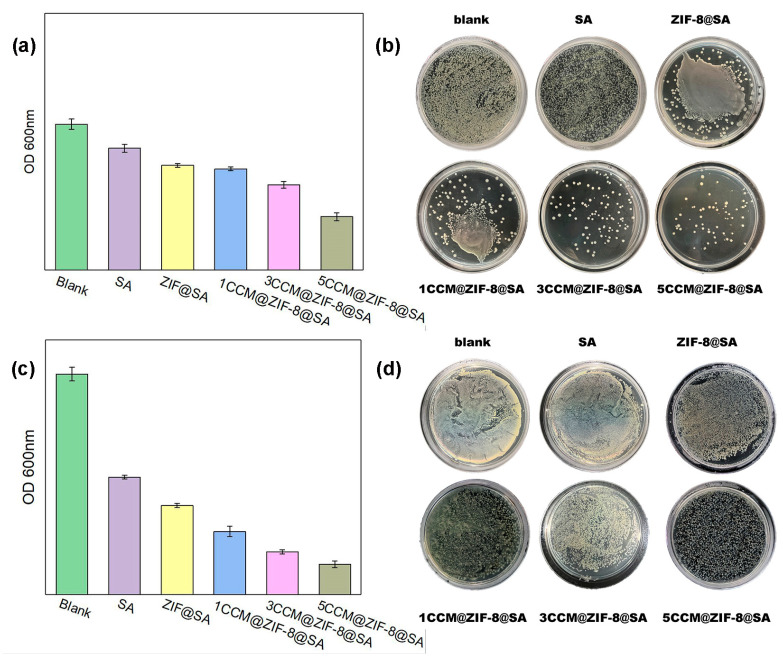
Quantitative measurement of *E. coli* (**a**), *S. aureus* (**c**) survival after 24 h of incubation, bacterial inhibition ability of the SA, ZIF-8@SA, and 1/3/5CCM@ZIF-8@SA against the clinically established bacterial pathogens [*E. coli* (**b**) and *S. aureus* (**d**)].

**Figure 7 ijms-24-11439-f007:**
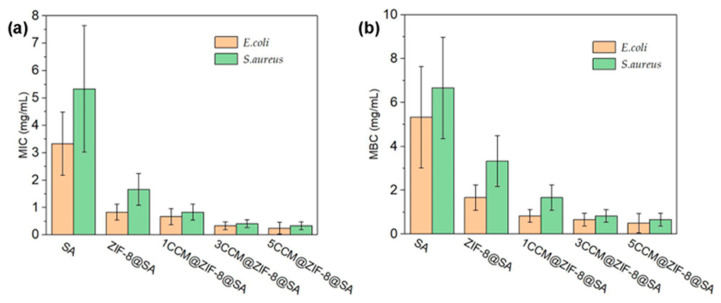
(**a**) MIC values and (**b**) MBC values of SA, ZIF8@SA, and 1/3/5CCM@ZIF-8@SA against *E. coli* and *S. aureus*.

**Table 1 ijms-24-11439-t001:** Surface area, pore volume, and average pore diameter of ZIF-8 and 1/3/5CCM@ZIF-8.

	S_BET_ (m^2^/g)	V_P_ (cm^3^/g)	Pore Width (nm)
ZIF-8	968.3	0.461	3.302
1CCM@ZIF-8	951.9	0.446	2.907
3CCM@ZIF-8	949.1	0.388	2.897
5CCM@ZIF-8	876.2	0.315	2.813

**Table 2 ijms-24-11439-t002:** Sample quality and CCM loading volume.

Materials	SA	ZIF-8@SA	1CCM@ZIF-8@SA	3CCM@ZIF-8@SA	5CCM@ZIF-8@SA
Quality	0.0396 (±0.002) g	0.0462 (±0.004) g	0.0465 (±0.002) g	0.0471 (±0.006) g	0.0485 (±0.009) g
CCM loading	0	0	0.0003 g	0.0009 g	0.0013 g
CCM release (72 h)	0	0	0.0492 μg	0.199 μg	0.332 μg

## Data Availability

The data that support the findings of this study are available from the corresponding author upon reasonable request.

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
