# Peer review of "Flexible Curcumin-Loaded Zn-MOF Hydrogel for Long-Term Drug Release and Antibacterial Activities"

_ijms, 2023, doi:10.3390/ijms241411439_

Round 1

Reviewer 1 Report

The manuscript entitled "Flexible curcumin-loaded Zn-MOF hydrogel for long-term drug release and antibacterial activities" prepared by Wang Lu-Ning et al. describes the development of a flexible hydrogel composed of sodium alginate (SA) matrix and CCM-loaded MOFs for long-term drug release and antibacterial activity.

The authors used SEM, FT-IR, X-ray diffraction XRD, UV-Vis, Raman spectroscopy, and mechanical property tests to investigate the shape and physicochemical characteristics of composite hydrogels. The results demonstrated that the composite hydrogel was highly twistable and flexible, allowing it to mechanically conform to human skin. According to the authors, the as-prepared hydrogel could capture efficient CCM for slow drug release as well as effective killing of bacteria.

The paper appears to be interesting, but I feel it requires substantial modifications and further examinations. Only then will it be accepted for publishing in the International Journal of Medical Sciences.

Here are some of my comments and recommendations for authors:

1. The paper's introduction should be improved by demonstrating what has previously been published in the literature on related topics as well as what is original and novel in this study.

2. What is new in the reviewed paper compared to the article described by Abdul Jabbar et al. ("Improving curcumin bactericidal potential against multi-drug resistant bacteria via its loading in polydopamine coated zinc-based metal-organic frameworks", Drug Delivery2023, 30, 1, 2159587), aside from replacing polydopamine (PDA) polymer with sodium alginate (SA) in the reviewed manuscript?

3. Please increase the font size in Figure 1a; it is absolutely invisible in this image.

4. It is controversial that the authors proved the successful incorporation of CCM into ZIF-8 by seeing just a blue shift from 3490 to 3435 cm-1 in the FT-IR spectrum. What about other distinctive bands, such as at 1626 cm-1 (C=O of CCM)? Have they shifted in any way? I request a detailed examination of the FT-IR spectra of the generated material's.

5. The existence of ZIF-8 reflections on the spectra of ZIF-8@SA and 5CCM@ZIF-8@SA appears to be highly questionable. The ones displayed in the image are nearly invisible, which may suggest an absence of ZIF-8 in the resulting material. Please repeat the test.

6. Please perform the following tests to thoroughly characterise the obtained material: TGA analysis to determine thermal stability and estimate solvent-accessible pore volume; adequate stability of the material and perhaps nitrogen adsorption and desorption isotherms to determine the textured parameters of the material.

7. Enlarge the fonts in Figure 4: a, b, and c. Please indicate the cumulative CCM release in percentage. The figure 4c indicating 95% CCM release in 24 h appears to be very questionable when compared to the quantity of CCM released from the other samples (approximately 35% and 27% from 3CCM@ZIF-8@SA and 5CCM@ZIF-8@SA, respectively)?

8. In my opinion, the authors should additionally conduct studies of minimum inhibitory concentration (MIC) for each sample, which is defined as the lowest quantity that inhibits detectable bacterial growth.

 Moderate editing of English language required.

Reviewer 2 Report

The work presented by Li Jiaxin has a good concept but suffer to some problems. Scientifically, the work is lacking important characterizations, some results are misleading, figures are too small and the statistical analysis is missing. Below point to point review is reported.

Line 75: what the ZIF-8 acronym does it means?

Line 94: the description of preparation of ZIF-8 and ZIF-8@SA are missing.

Line 95: how the optical images were recorded?

Line 159: there are some issues about the release studies. First, curcumin is a very insoluble substance in water but especially at neutral and alkaline pH has a rapid kinetic degradation. For these reasons it is necessary to justify how it is possible to obtain the results reported in the manuscript. Did a sort of solubilizing agents used? how is it possible that curcumin has remained stable for up to 72 hours? Furthermore, without a quantitative analysis of the samples it is impossible to understand the actual percentage of curcumin released. This part needs to be deep overhauled please check this: https://doi.org/10.3390/biomedicines8100425.

Line 173: how was the percentage of curcumin released calculated if no quantitative analysis of hydrogels was carried out?

Line 232: please add the acronym for methanol in the material section.

Line 222: this paragraph must be improved. First the description of the hydrogel and final formulation is hard to read and misleading. Then, the SA solution has been entirely added or in aliquots? in which methanol solution? It is not clear how the particles are prepared and if the curcumin was dissolved in the same solution. Then, how was the reaction checked to make sure it was complete? Did the hydrogel dry in some ways? hydrogels have been weighed after preparation? it is essential to know whether the samples are homogeneous and reproducible.

Line 238: a crucial characterization is missing. Considering the high instability of curcumin in water, it is important to perform a quantitative analysis on hydrogels in order to determine the effective amount of encapsulated active. In this way it is possible to determine that all the curcumin used is actually inside the hydrogel, which has not degraded. This information is essential for the validity and reproducibility of the preparation method but also for subsequent release and microbiological analyses. Please, take into account to carry out this evaluation.

Line 255: how the area of hydrogels was calculated?

Line 260: how it is possible diluting a solution to reach a concentration of 0 μg/mL? is it a typo? Please check.

Line 264: please add how the release data are expressed.

Line 265: what is the amount of hydrogels used for the release studies?

In the point of view of English language checking is necessary, lot of sentences are hard to read and full of typos

Round 2

Reviewer 1 Report

Dear Authors,

Thank you so much for taking the time to provide responses to my suggestions for improvement.

I would like to note out that the FT-IR spectra contains bands rather than peaks (correct the entire text of the manuscript).

Furthermore, I propose that the authors add the work of Abdul Jabbar et al. to the list of references ("Improving curcumin bactericidal potential against multi-drug resistant bacteria via its loading in polydopamine coated zinc-based metal-organic frameworks", Drug Delivery 2023, 30, 1, 2159587) and indicate in the introduction section how their work differs from that which they cited and what is innovative in it.

Minor editing of English language required.

Reviewer 2 Report

I appreciate that the authors have massively modified the manuscript based on the comments received. However, there are still parts that need to be fixed.

In the description of ZIF-8, ZIF-8@SA and 1/3/5CCM@ZIF-8@SA preparation method the model and manufacturers of the instrument used are missing. Furthermore, some sentences are misleading in terms of syntax and hard to read.

Drug release part remained with criticisms. First the description of curcumin quantitative analysis is missing in the material and methods section. Furthermore, How the new release percentage have been calculated? for example for 1CCM@ZIF-8@SA it has gone from 95% to 11.4% and according to quantitative analysis where the percentage of CUR present in the sample is 50% the percentage of release should have been halved and not reduced by 10 times. Moreover, these release experiments suffer from a serious criticality, curcumin is practically insoluble in water and the concentration gradient and therefore the sink conditions are difficult to maintain. Usually, when release studies have been performed under optimal conditions, a plateau is observed when the formulation runs out, as the concentration gradient is lost. In this case, however, a plateau is appreciated although within the formulation is still present most of curcumin. This phenomenon is due to the low solubility of the substance in the acceptor fluid, and it is therefore imperative to repeat them using a medium more suitable for curcumin.

In the description of concentration range for the Curcumin calibration curve the 0 µg/ml as it refers to the pure methanol should be removed.

In conclusion, statistical analysis is still missing.

The syntax must be checked as some sentences are difficult to read.

Round 3

Reviewer 2 Report

I would like to express my gratitude to the authors for their responsiveness to the comments they received and for their efforts in enhancing the manuscript with new experiments. I appreciate their attention to addressing the criticism regarding the release test, which is now more clearly explained, as well as the improved description of the hydrogels' preparation method. However, there are still some sentences that remain hard to read, and it would be beneficial to provide further clarity in describing the methods.

English should be checked both in their form, spell and syntax

Author Response

Thank you for your opinion, we have put the refining part in the attachment, please see the attachment.
